# Re-Membering Catholicity: Higher Education, Racial Justice, and the Spirituality of the Posthuman University

**Jeffrey S. Mayer**

Department of Theology and Religious Studies, Villanova University, Villanova, PA 19085, USA; jmayer04@villanova.edu

**Abstract:** In the reinscribing of white supremacy in the United States, the contemporary university as a place of exclusion presents a problem of religion. Approaching religion as "the search for depth" and addressing the "techno-myths" of betterment, longevity, and the rituals of enacting these myths that capture today's social imaginaries, this paper proposes an alternative to religious faith in "rising" and the rhetoric of the contemporary American technocratic-meritrocratic paradigm. Adopting the posthumanist methodologies of reflexivity and diffraction, the author argues for an embodied catholicity of the university as a community, an open system rather than a pre-formed locus to which racially minoritized students are "added" or "included". In advancing the co-creativity of a Catholic-pluriversal university via an ethic of love and care, the author presents a Christian spirituality that is itself a technology that offers the hope of enacting a more life-giving congruence between the sacred and the secular than the myth of Manifest Destiny and the racialized violence that is the continued manifestation of that mythos. Embodied in the posthuman mystic's practices of "re-memory," the author presents Christianity as a performative-pluralistic religion of evolution, one of common action with the potential to draw into something new the energies of creativity in today's university.

**Keywords:** catholicity; Catholic Social Tradition; critical race studies; decoloniality; evolution; higher education; mysticism; posthumanism; solidarity

## 1. Introduction and Methodology

"What if white America is only willing to provide pity and charity, but not justice?" (Yancy 2017, p. 120). As this essay will demonstrate, this question from critical race theorist George Yancy lands squarely at the feet of those of us in academia, particularly in Catholic institutions of higher education. As Christian ethicist James Keenan has noted, "as undergraduates progress in higher education, they become less interested, on average, in promoting racial understanding".[1] In a time when the "campus climate" around racial diversity is worsening, necessary are both Anthony Jack's affirmation, "Access alone is not enough for fostering inclusion",[2] and Keenan's encouragement to move students, faculties, and administrations to greater partnership and community (Keenan 2015, p. 156). Still, the societal roles played by colleges and universities, those institutions at the "animating heart of meritocratic aspiration" (Sandel 2020, p. 156), demand an affective re-imagination of the metaphysical assumptions underlying a narrow neoliberal vision of what the human person is.

Taking a cue from Keenan and Jack, *is* the university a community, and if so, how is community possible? What do we think a university is for, and *for whom?* Even more, *how* do we think of the university? The subsequent exploration of these questions relies upon what posthumanist thinker Rosi Braidotti has termed the posthumanities. The often-dismissed radical epistemologies of feminist, decolonial, and critical race studies must find "transdisciplinary cross-pollination across the board" with the life sciences, Braidotti advances. Guided by Felix Guattari's sense of *transversality* as an antidote to the corporatization of the university, Braidotti's posthumanities model a relationality and



affect in education, an understanding of person not as an autonomous rational individual but as a capacity *to affect and be affected*, and a "more complex assemblage that undoes the boundaries between inside and outside the self" (Braidotti 2019, pp. 45, 142–44). To this end, this study adopts the posthumanist methodologies of *reflexivity* and *diffraction*, by which we may "confuse and entangle the boundaries we impose on the world in order to make sense of that world" (Hayles 1999, pp. 8–9) and find "productive connections instead of limiting the analysis to a critical classification exercise" (Delio 2020, pp. 137–38). Posthumanism, as a mode of analysis skeptical of the centrality of the human in Western philosophy, opens our eyes to the very erasure of embodiment (Hayles 1999, p. 4) that is a feature common to the liberal humanist subject of today's techno-meritocratic religion. In learning from Katherine Hayles "how information lost its body," we see a certain conception of the human standing at the center of our current tyranny of merit, "a conception that may have applied, at best, to that fraction of humanity who had the wealth, power, and leisure to conceptualize themselves as autonomous beings exercising their will through individual agency and choice" (Hayles 1999, p. 286). With a marked emphasis on cognition at the expense of our embodiment, the grafting of minoritized students on to neoliberal centers of learning that call themselves "Catholic" invites a diffractive-reflexive re-engaging of the *catholicity* of such institutions.

This paper will argue for an embodied catholicity of the university as an open system rather than as a pre-formed locus to which racially minoritized students are "added" (i.e., admitted) or even "included." In entering with each other into the creation of spaces of vulnerability and care, such spatio-temporal entanglements begin *within* and invite a movement *without*, as today, we risk engaging the memory of our racialized past in the United States and enact a future that presses beyond universities as places of inclusion and tolerance toward ones that embody consciousness of the whole. Rather than the "damaged human nature that is liberal subjectivity" (Kroker 2012, p. 119) of our techno-meritocratic age, an incarnational religion holds the potential of awakening our co-creative capacities as hybridized persons in loving solidarity within and from the triadic spaces of our universities.

## 2. The Phenomenon of the University: A Posthuman Re-Imagining

The death-dealing violence of white supremacy in this country and its reinscribing within our academic halls as places of exclusion present us with a problem of religion. Today's "techno-myths" of betterment, longevity, "and the rituals of purchasing the technological means of enacting these myths" capture our social imaginaries, and, as Ilia Delio contends, in many ways, serve as a religious ethos unto themselves (Delio 2020, p. 135). A corresponding faith in "rising" and the rhetoric of opportunity ritualize what philosopher Michael Sandel calls our contemporary American technocratic-meritrocratic paradigm (Sandel 2020, pp. 22–23, 85, 104). The loci of this discussion are the very admissions offices and classrooms of the university.

Today's university can be an isolating place, a prime locus for the techno-myths and rituals of our increasingly transhumanist age. Flying in the face of a lingering mechanistic model of reality from the early modern period, evidence for the primacy of a *relational* ontology challenges those metaphysical assumptions underlying the contemporary university.[3] "While there is much good that takes place at universities today", Delio argues, "still, the whole process of higher education points to a fragile future", for "the university is a 'multiversity' of sub-specialized disciplines that does not promote an ecology of the mind but a mechanized system of information" (Delio 2013, p. xxiii). Such "parceling out the whole into specialized fragments" bears striking resemblance to the critique of James Keenan on the "siloing" of the university and those structures that "keep the university fragmented without an underlying structure or foundation that relates one fiefdom or silo to another" (Keenan 2015, p. 35). This lingering organization of the university into *Wissenschaften* or academic disciplines, originally premised on the adoption of a Cartesian model of pure objectivity, process theologian John Cobb argues, leaves "no place to consider

how subjective experience could play a role in determining what happens".[4] The means for sustaining this institutional status quo[5] further reinscribes this prevailing institutional epistemology, grounded in an ontology that imagines faculty, students, and administrators as individual, atomistic entities, concerned primarily with doing "one's work" in private.

It is no wonder, then, that racially minoritized students[6], even after being admitted to college, report feeling disconnected. As sociologist Anthony Jack reports, this is especially true for the "privileged poor" and "doubly disadvantaged" students.[7] Further uncovering the "false construction of the corporate university as set apart from real life," bell hooks unveils this fragmentation for what it is. She pronounces, "For many students from backgrounds that are marginalized by race, class, geography, sexual preference, or some combination, college continues to be a place of disconnection" (hooks 2003, pp. 41, 177). However, what causes racial minoritization and this disconnection? What if universities are always already spaces of racial marginalization which then further reinscribe a minoritizing of those who are not "white?"[8] While colleges and universities in recent decades have advanced the recruiting of African American and Latinx students, Sandel argues that little has been done to increase the proportion of lower income students, who are disproportionally people of color. Per a meritocratic faith, expanding access to higher education has become the "common sense" response to economic inequality in recent decades. Still, studies such as the "Mobility Report Card" paint a less optimistic picture. Examining the economic trajectory of 30 million college students from 1800 colleges during the period 1999 to 2013, Raj Chetty's team found that the mobility rate is especially low at private colleges, which enroll fewer poor students and where only about 1 student in 10 rises even 2 quintiles on the income ladder. Contrary to the American religion of "common sense" and its meritocratic faith, colleges and universities do less to expand opportunity than to consolidate privilege. As Sandel asserts, "American higher education is like an elevator in a building that most people enter on the top floor" (Sandel 2020, pp. 169–70).

At this site of the college campus that reinscribes the meritocratic rituals of the great "sorting machine" (Sandel 2020, p. 155) of the neoliberal mythos of America, we turn to the posthumanist dialogical partners of New Materialism and decoloniality. Entangled with the governing ethos of the autonomous Cartesian subject, that legacy of the Enlightenment and faith in unbridled technological progress marks our contemporary collective imagination. From aspirations of overcoming disease to digital immortality through brain downloading, the accepted "givenness" of the metanarrative of transhumanism and its autonomous liberal subject that recapitulates a reduction of personhood to enhancement, rationality, and Godlike power[9] invites an entangling of a new discourse of *embodiment* within the technocratic universities in which the myth of merit reinscribes privilege and power. The term "New Materialism," a term coined by Manuel DeLanda and Rosi Braidotti during the 1990s, works to show "how the mind is always already material (the mind is an idea of the body), how matter is necessarily something of the mind (the mind has the body as its subject), and how nature and culture are always 'naturecultures'" (Dolphijn and van der Tuin 2012, p. 48). In her book *Meeting the Universe Halfway*, Karen Barad has especially taken the last point in the above triad to develop her diffractive method.[10] In drawing upon the philosophy of early quantum physicist Niels Bohr, she exposes the dualist episteme that continues to structure the academy (and beyond). Significantly for the current discussion, this diffractive method exposes an epistemological re-producing tendency in our common philosophical practice of "reflection," namely, "that the cordoning off of concerns into separate domains (matters of 'fact' for the sciences and 'concern' for the humanities)" most often governs the way we think about our siloed work in the academy. Crucially, for Barad, *agency* is not only a "human" affair. In practice, she argues, it's an anthropocentric view that undergirds the cordoning off of disciplines. In that even the mind itself is matter, *all* matter is agential—it *is* because it relates. From subatomic particles to ecosystems (including the ecosystem of the university), there are no monadic entities.

Rather than partitioning what "is" to the sciences ("reality") and what "matters" to the humanities ("meaning"), Barad advances an "ethico-onto-epistemology" that is operative in the current discussion. As she asserts, "matter and meaning cannot be severed" (Dolphijn and van der Tuin 2012, p. 69). Here, in considering the *afterlife of slavery* on Black bodies,[11] the ethical implications of Barad's diffractive emphasis on differentiating comes to the fore: "Differentiating is not about radical exteriorities but rather what I call agential separability". It's not about "othering", Barad notes, "but on the contrary about making connections and commitments". In other words, in pointing to the ethical at the nexus of the ontological and epistemological, "the very nature of materiality itself is an *entanglement* . . . Ethics is therefore not about right responses to a radically exteriorized other, but about responsibility and accountability for the lively relationalities of *becoming*, of which we are a part".[12] Needed, then, is a re-imagining of our overly facile sense of racial inclusion in our universities today via a questioning of the very metaphysical assumptions underlying how we do what we do in the academy. The dualistic mind that so often pits scientific realism against social constructivism is itself, as Barad shows, caught in a feedback loop premised on the metaphysical presupposition that beings exist as individuals. From this epistemological starting point of *representationalism*—the belief in the ontological distinction between representations and that which they purport to represent—that which is represented is held to be independent of all practices of representing (Barad 2007, p. 46). Rather, via Barad's philosophy of agential realism, *performativity*—"knowing as an ongoing performance of the world"—becomes our *making* real in our enacting of reality itself (Barad 2007, p. 149). In an agential realist account of reality, as an "apparatus", the university appears as an agent, not as a reductionistically corporate "given," and, significantly, as more than a mere collection of individuals. The university becomes a "material-discursive practice", and as an apparatus, we can begin to view the university as more of a *phenomenon* than a "thing", and as an "open-ended practice involving specific intra-actions of humans and nonhumans".[13]

Here, as a conversation partner, decoloniality lays the condition for the possibility of interrogating the reality of the university.[14] Is the university a "given" institution that faculty, staff, and, significantly, students are beholden to as already pre-formed? Taking a cue from Barad for the co-constituted nature of reality as always becoming, a dynamic intra-activity via performativity, the challenge of a decolonial reading of the university in its American context answers a resounding "No!" As the scholar of Christian cosmology Thomas Berry contends, today's universities stand in the dubious tradition of the "Manifest Destiny" complex. Codified in the very origins of the United States as a country, the "self-exaltation" found in the Constitution "was the fulfillment of the anthropocentric ideals of Western civilization" (Berry 2009, p. 154). The inseparable interweaving of social, political, economic, and religious factors, the complex of the sacred-secular alliance of Catholic missionaries and the U.S. government continues in a legacy of, Berry critiques, "deep anxiety in approaching other people or modes of being". The academy, existing in a particular place within the "white authority structure" (Tinker 1993, p. 117), perpetuates the isolating of relationships when we, whose voices come primarily from within, are speaking in our silos. While the company may be good, we must question: at whose cost?

Through their collaboration, Catherine Walsh and Walter Mignolo offer a further entangled de-and re-constructive reading of the analytic of the *coloniality of power* and the possibilities for de-linking from the "promises of modernity". As Walsh articulates, it is in the "cracks and fissures" of the 500-year long project of that model of world power called "coloniality" that the "de-monopolizing" of life is possible.[15] Nothing less is at stake than the invention of the "New World", an entangled product *and* perpetuator of the universalizing onto-epistemological domination of white, Christian colonizers in the place *Abya Yala*.[16] Thus, before domination was expressed (ethically) in violent praxis against "othered" peoples (particularly native peoples of this landmass and enslaved Black bodies), it was the relegating of non-modern/non-white (Christian) cosmologies and the very colonization of space and time in the early modern period that co-constituted

modernity/coloniality.[17] A crucial intra-active place of discursive possibility between posthumanist and decolonial thinkers is the notion of discourse itself, for to question the givenness of a technocratic "progress" and the contemporary university as the inheritor and purveyor of the givenness of a certain autonomous agency and choice (ala Hayles above), the very question of how the university is entangled in this history and a wider cosmological understanding of reality is needed.[18]

What might it look like to take a topological read of the contemporary American university as "entangled?" If, as agential realism advances, phenomena are the smallest material "units" of reality, and if matter itself does not refer to an inherent, fixed property of abstract, independently existing "objects," then the university becomes a material-discursive practice. A "reconfiguration of the world through which the determination of boundaries, properties, and meanings is differentially enacted" (Barad 2007, chp. 4, esp. 151), the university may be thought of as a *body*, for the material and discursive as they are mutually implicated in the dynamics of intra-activity complicate any facile sense of the university as an already pre-formed "given". Thus, the very phenomena that take place in our universities are not happening in a "space . . . a collection of pre-existing points set out in a fixed geometry, a container as it were for matter to inhabit" but are always already co-constitutive/iterative of the "changing topology"[19] that is the material-discursive practice of the university. The universities that promulgated the international law that legitimized the imperial appropriation of land and propagated the epistemic social classification of racism were themselves agentially co-constituted by the same epistemic invention that is "America".[20]

### 3. Re-Membering the Racialized Past: The Challenge and Hope of a Posthuman Mysticism

To re-imagine the university topologically—as enfolded and, therefore, agentially responsible for the past that created it *and* radically open to an unwritten future which it itself co-creates—is to disrupt the common narrative of an "unfolding of universal history" via a particular understanding of the human person—the Cartesian subject. It is the work of *re-memory*, "putting back together parts that have lost touch with one another and reaching out toward a complexity too unruly to fit into disembodied ones and zeros" (Hayles 1999, p. 13). At work here is a posthuman imagination, more humanizing than the kind of techno- meritocratic discourse premised on an ontologically monadic individual that has formed and that has been informed by a public imaginary of the purpose and the ends of education, that is, credentialism and social esteem, the "hubristic conceit" of higher education today (Sandel 2020, p. 151). To see ourselves as entangled in the re-membered, posthuman university is to open those of us who are white, Euro-Christians to re-imagine ourselves in a contemplative move that may allow a more loving look at our realities as entangled with those of our racially minoritized students.[21]

If, as Karen Barad's agential realism challenges, "The past is never finished once and for all and out of sight may be out of reach but not necessarily out of touch" (Barad 2007, p. 394), then what does the ongoing effects of the afterlife of coloniality for young people of color, generally, and, particularly, of slavery for young Black people in the United States say about the "catholicity" of today's university? Recent studies on the state of higher education reveal the reality that a "dual system of racially separate and unequal institutions" has developed in the U.S.; one that is inherited from inequities in both elementary and secondary education (Carnevale and Strohl 2013, pp. 7–54). Complicit are the very colleges and universities that, while touting "diversity" as a good in theory, reproduce systemic white racial privilege across generations. While the overall share of American undergrads in recent decades has shown an increase of African American and Hispanic representation, simultaneously, new white student enrollments have gone toward the top 468 most *selective* colleges, with Black and brown students "left behind" in open-access, two- and four-year schools.[22] As the work of Anthony P. Carnevale and Jeff Strohl reveals, racial polarization in higher education matters because *resources* matter.[23]

As more selective institutions provide considerably more resources per student on average, even well-intentioned institutions of higher education can extol an increase in minoritized attendees while in practice perpetuating systemic inequality based on race.

Contra an overly facile "march of progress" narrative of modernity/coloniality and the "banality of slavery to the university and the social world that it was so crucial in maintaining" (Boggs and Mitchell 2018, p. 451), an honest recognition is needed on the part of Catholic universities that the racialized present is one that those same universities, even unwittingly, contribute to in this country. An (often unstated) expectation that the "privileged poor" prove their worth via "politics of respectability"[24] reveals that even in our "inclusive" and "diverse" institutions, the afterlife of slavery is alive and well. Such a "cruel optimism"[25] for a delimited Black selfhood conflated with whiteness begs the question: What does it mean to call ourselves "Catholic?" When an ideology of "color-blindness" dominates the discourse in our institutions that continue to be overwhelmingly controlled by white people, a strategy of "niceness"[26] becomes a facile response to diversity, further perpetuating a culture of racism, one of "natal alienation"[27] entangled in living memory with the ghosts of slave, sharecropper, and lynched ancestors. Via the "adaptive character of whiteness",[28] our institutions of Catholic higher education that exist today are entangled in white supremacy. A diffractive, posthuman agential-realist reading of our Catholic institutions offers another way beyond the safe harbors of extolling "diversity" and "inclusion." A topological reading of the university, entangled in the colonial project of "America", demands that Catholic universities recognize that the afterlife of slavery on Black bodies shares an iterative history with our religious institutions.

Paradoxically, in honestly reconning with the role of religious institutions as purveyors of such violence, the *hope* of religion as the search for depth[29] and as a cultural vehicle for discerning the shape of institutional agency today can come to the fore. Working with the posthumanist thought of Donna Haraway, Ilia Delio adds a clarifying voice in her critique of the place of religion in an era when many of us "think of religion as something institutional, static, and ritualistic" (Delio 2013, p. 73). Rather, she argues, from the Latin root meaning of the word (*ligare*, "to bind back"), religion is at the dynamic heart of the process that is evolution. "Religion brings to light the transcendent nature of the cosmos," Delio articulates, "the excess of love that lures the whole into more love and a new future" (Delio 2013, p. 205). The cosmology of the Jesuit priest-scientist Pierre Teilhard de Chardin challenges us to speak of spirituality as performative religion, for it is only a meeting of religious traditions on the level of mysticism and common action that religion as a "new tethering" to the whole is possible. The hope of this convergence of religions and the development of an interspiritual religious consciousness that may bind us together lies in the person as posthuman mystic, "the one who has the courage to live in the God-ing moment, connecting and creating the art of life" (Delio 2020, pp. 163, 165, 200). In the pluripotentiality of the hybridized nature of the posthuman mystic as a *cyborg*, one who inhabits more than one world and is thus a boundary transgressor who is therefore about "dangerous possibilities," we witness the posthuman as a "new type of person emerging in evolution" (Delio 2020, pp. 106, 131). Not only is the university a discursive process, but, more importantly, *we* are an "ongoing creative process of unknown future" (Delio 2020, p. 107).

Approaching the question, "What do we do with this now?" regarding the Christian religion's complicity, historical and contemporary, in anti-Black racism invites the posthuman mystic into the heart of the conversation. It is a conversation about *conversion*, not just individual but institutional as well. In her framing the transgressive actions of Henriette Delille to start a religious congregation for free women of color in antebellum Louisiana, theologian Shawn Copeland gives shape to this posthuman, cyborgian mysticism. Delille's disruption of the morally ambiguous strategy for freedom, the "institutional arrangement" of *plaçage*, required a break with her cultural and religious horizon or worldview.[30] Copeland's terming this break as "conversion", in the Lonerganian sense of the word, resonates with the centrality of *interiority* in a Teilhardian theological cosmology.

"Religious practices", Delio asserts, "should connect inner and outer worlds into a unified whole, so that what fills our inner worlds, our hearts and minds, is expressed in our outer worlds, our actions and decisions". As in a diffractive topological reading of the university a la Barad, "even what is 'inside' and what is 'outside' are intrinsically indeterminate" (Barad 2007, p. 161). Copeland's theological reading of Delille demonstrates the fruit that comes with a critical and, I would argue, *contemplative* reflection on what could (and has often been) an uncritical "celebration" of the religious foundations of our Catholic universities that, seemingly as in a perfect cause-and-effect, linear path, have sprung from the fertile American soil. For Copeland, it is a *liberative* theology of religion that underscores and makes possible an honest confronting of the racist history of the Catholic Church in the United States. Through a womanist lens, the "binding back" of religion counteracts a tendency toward religiosity, that reduction of religion that results, Delio argues, "when God is absolutized by constructing foundations that are purported to be immutable, when fixed structures cannot adapt to changing circumstances" (Delio 2013, p. 113).

As Henriette Delille demonstrated in the founding of an order of Black and multiracial Sisters, boundary crossing has political consequences (Delio 2020, p. 106). Participating in the socio-political structures of this country, our universities themselves are in need of conversion via the boundary-crossing practices of a *love* ethic among administrators, faculty, and staff who, in our silos, often struggle in even imagining what a genuine relationship may look like with students feeling the marginalization of white supremacist social structures. In contemplatively challenging our own already-entangled agential responsibilities in white-dominated American institutions, paradoxically, the revelation emerges of the "subversive power of love". The hope of a posthuman mysticism of the Catholic academy lies in the *God who is incarnate,* self-engaged in the world in the Christ, a "symbol of that capacity of every person and every creature to be united in love" (Delio 2020, pp. 173, 181). Knowing contemplatively, engaging the Christian *practice* of re-memory in stories of Black mystical love such as Delille's, is a crucial component of a conversion that joins the inner and outer worlds of individual members and our Catholic institutions. To do so, as Copeland reflects, is to incarnate the dangerous memory of the crucified Jesus.[31]

With their roots in culturally disparate structures that isolate and form young people to be "competitive in the marketplace", a deep conversion is demanded for genuine relationships to grow in order for community to be possible in today's Catholic universities. Conversion means a willingness to transgress public–private boundaries and to be transformed by a "pedagogy that refuses to separate individual problems and experience from public issues and social considerations" (Giroux 2014, p. 46). Integrating this pedagogical way-of-being *exteriorly* throughout one's work, flowing from *interiorly* embracing the imperative as a teacher and fellow journeyer with young people experiencing dislocation, is to make a living religion of our nominally religious institutions as "networks of performance" (Delio 2020, p. 198). What is needed today from within the academy is something more than a pastiche of tolerance trainings and team-building exercises that become another commodity for the compositional-majority white students who continue to occupy the vast majority of the desks in our "selective" institutions. What is needed is a performative, pluralistic religion of common action. Without an inner–outer ethico-onto-epistemology grounding the intra-activity of what we do *across* the Catholic academy, solidarity will remain a very quaint-sounding word in our mission statements, a final topic to which we now turn.

## 4. Co-Creating from the Splice: Toward a Catholic-Pluriversal University

What does it mean to be a person? The re-imagining of our categories of personhood point to the need for a renewed sense of *catholicity*, of faith-making-wholes to deepen our capacity for empathy across racial lines. As Rosi Braidotti prompts, personhood is a capacity to affect and be affected. To be a person, Ilia Delio suggests, is to be "a conscious being in relation to everything else, one in whom the matrix of relational life is expressed in a particular way and who contributes to the unfolding of the world in a particular

way" (Delio 2020, pp. 24–25). The posthuman opens the possibility of "new ways of thinking about what being human means", in what Katherine Hayles conceptualizes as a shift from a "presence/absence" paradigm to one of "pattern/randomness", a dialectic in which "meaning is not front-loaded into the system" and the system does not proceed along a trajectory to a known end but instead evolves "toward an open future marked by contingency and unpredictability" (Hayles 1999, p. 285).

What does this mean for today's universities, especially those that call themselves Catholic? How might the techno-meritocratic educational ends of social esteem and credentialism give way to the posthuman university as an open system? Here, the language of the Catholic Social Tradition in the voice of Pope Francis issues a further call to embody the conversion demanded of re-memory. While today the "undifferentiated and one-dimensional" technocratic paradigm "exalts the concept of a subject who, using logical and rational procedures, progressively approaches and gains control over an external object", Francis laments an "authentic humanity calling for a new synthesis" goes "almost unnoticed" (Francis 2015, nos. 106, 112). What is needed is a "renewal of humanity itself", a "distinctive way of looking at things, a way of thinking, policies, and educational program, a lifestyle and a *spirituality* which together generate a resistance to the assault of the technocratic paradigm".[32] This entails the consciousness of *depth* in life: a living religion that liberates us to live *from the inside-out*, for only from the inside may we see that *solidarity* is the very heart of our university missions.[33] Amid the complexifying network of interdependent relations via an emergent consciousness of a global common good, such is the possibility of an affective community's love relations (Mayer 2019).

Here is precisely where an incarnationally living religion for our Catholic universities, a posthumanist spirituality, may shape our affections as teachers and administrators. To co-create the common good via the intra-activity of a community of love is to act, as Hayles terms it, "from the splice". This is an envisioning of a different type of human, one whose identity, as Delio interprets, "is not a given but an ongoing creative process, a cybernetic loop between the interspace of information and the core experience of personal existence".[34] The academy, microcosm of larger complex systems as we saw above, is itself a "splice": a space in which the human intra-acts with a wider system of relationality. This *deep relationality* is often obscured in our quotidian being, given our dualistic thinking and binary logic. The hope of a contemplative, affective religion of action lies in its inherently triadic logic, "a progressive evolutionary way of learning" in which the "intermediate complex mediates the relationship between the 'same' and the 'other'" (Delio 2020, p. 127). Rather than the "either-or" paradigm of the university as producer/disseminator of knowledge, or a more recent one of producer/disseminator of market-oriented credentials, a triadic reading of the university as complex dynamical system can place the re-membered Black bodies of racialized violence at the heart of the material-discursive practices of our universities. Embodying a common good beyond a pre-formed end, the triadic entanglement of the "university—person—public" holds the hope of co-creating a *depth* dimension in asking for whom and to what ends the academy exists.

Living from the inside-out means that to be concerned with the public and common good, the stated values of the university must start within. For the Catholic university, this means actually embodying the principles of the Catholic Social Tradition. Donna Haraway's heuristic of hybridity further exposes the pathology of the liberal subject in pointing to the person as already a *techne* of a larger *natureculture*, a pluripotential centration of energies to make something new.[35] To our own detriment, the techno-meritocratic pathology of the credential-seeking individual is reproduced in our university admissions policies. As Michael Sandel argues, while public debate has raged over affirmative action for racially minoritized students, "colleges have quietly practiced what amounts to affirmative action for the wealthy" (Sandel 2020, p. 170). From legacy students to athletic scholarships, on the whole, applicants who benefit from admissions preferences are disproportionately wealthy and white. Revealing such "meritocratic sorting" (Sandel 2020, p. 172), the specter of our larger neoliberal techno-myth of "betterment" points to the painful truth

that, while colleges celebrate diversity, an ethos of perfectibility determines who gets ahead in American society in tandem with transhumanist aspirations. The "undifferentiated and one-dimensional" technocratic paradigm thus rears its ugly head in the very ableism of our purportedly inclusive academic communities.

Living from within the contingency of "pattern/randomness", a mystically embodied anti-racist catholicity engages the teacher as a hybridized-cyborg, intra-acting from and within the *natureculture* that is the phenomenon of her university. To co-create these pedagogical practices of a living religion, to live from the inside-out, demands new institutional practices of care. To that end, we might begin with faculty assessment practices. In reviewing the rank and tenure structures that demand our "multitasking" and often prevent a way of care from taking root (Noddings 2002, pp. 27, 36), what if, instead of meriting siloed, individualistic scholarly practices, administration teams created and implemented value-oriented assessments that account more for "non-academic" criteria, such as service?[36] Rather than re-present/re-produce faculty members who understand themselves as "free agents, independent operators", this posthumanist extension of agency across the siloed domains of the academy could co-create "service scholars," beholden to the institution as a whole. In this, we might live into and from a deep relationality, co-creating a common good "by building solidarity with staff and students, in order to ensure that *all* of us will have a future to share".[37]

"Institutions are also us," educational theorist Parker Palmer often notes.[38] As "sites of occlusion" (Yancy 2017, p. 8), a posthumanist agential realism discloses that we, especially Euro-Christian members of the academy, are already entangled in the ongoing *spacetimemattering* of our universities; a process whose performance is conditioned but not determined by the afterlife of slavery in the making of "America." In particular, a spirituality of our posthuman Catholic universities, a "consciousness of God in the splice" (Delio 2020, p. 186), calls for a sense of *vulnerability*, a kenotic openness to asking questions of ourselves that might move us to intra-act *with* our racially minoritized students, an *otherwise*[39] rather than the "promises" of modernity. In opening ourselves to performing the Catholic university as entangled in the *naturecultures* from which it was born and in which it continues to co-create, solidarity becomes an intra-active opening of our own interior spaces of vulnerability to exteriorly do so with each other. Here we embrace a "shared transience",[40] not of a "universal humanistic" sort, but authentically and nonanthropocentrically, inviting a cosmic living memory of this space *Abya Yala* that we continue to colonize and ontologize as "America." A spirituality of the posthuman university opens this possibility, with the posthumanities leading the way. The often-dismissed radical epistemologies of feminist, decolonial, and critical race studies that this study has employed must find "transdisciplinary cross-pollination across the board" with the life sciences. The posthumanities model a relationality and affect in education, an understanding of person not as autonomous rational individual but as a capacity *to affect and be affected*, a "more complex assemblage that undoes the boundaries between inside and outside the self" (Braidotti 2019, pp. 45, 142–44). In ritualizing acts of vulnerability from the admissions office to the classroom and lab a spirituality of posthuman Catholic higher education can become performatively real in our transformative practices within co-created spaces of care: ones of empathy, healing, and radical hope.

## 5. Conclusions

Living affectively in the splice is a contemplative matter. To approach George Yancy's opening question to us—"What if white America is only willing to provide pity and charity, but not justice?"—with our own sense of *authenticity* invites an uncovering of the contemplative heart of our Catholic centers of higher education. It entails the Christian *practice* of memory as constitutive of performing a university open to the future rather than beholden to a pre-formed past. Today's "grafting on" of Black students prompts a critical engagement of our category of "religion" that has been (de)formed through the Church's agential role in the project of coloniality. In our white educational spaces, in

which even our "nice" white students are largely viewed as "racially neutral" (Yancy 2017, p. 8), a *catholicity*, a consciousness of the whole, that can authentically claim itself as such must be grounded in a vulnerable, that is to say *kenotic*, embodiment of the Jesus who discloses the God of Love. In this kenotic intra-activity, in which Christ is continuously born, this universality must necessarily be *pluriversal* in expression, for, as Karen Barad's agential realism reminds us: "Differentiating is not about radical exteriorities . . . It's not about othering, but on the contrary about making connections and commitments." What are those (dis)connections today in Catholic higher education, and more to the point, *for and with whom do we make our commitments*? As phenomena themselves rather than ontological givens, as spaces still-coming-to-be topologically within the process we call *spacetimemattering*, our Catholic universities are both co-creators of and sites of resistance to the agentially delimiting discourse of white normativity. What we now have before us is an invitation to co-create practices of genuine *mutuality*,[41] in which "diversity" is not extolled in a cheapened way by pointing to our overall increase in accepted racially minoritized students. Amid our privatized and siloed work, we need to seriously consider the question: "Are you being as vulnerable with your students as you're asking of them?".[42]

What may emerge from the splice of the university today from those cracks and fissures of a colonial history that is not just past? The dangerous memory of the crucified Jesus and the danger of re-membering the racial past of our own institutional foundations in this country are inextricably interwoven within webs of whiteness. With great potential for good and for ill, the ambiguity of the human person is a truth that shapes our politics and sense—or lack thereof—of what our relation is to each other. Pluripotentially more humanizing than the bankrupt cultural construct of the neoliberal human subject, a mystical posthuman imagining of the university as a collective body reflects a view of the human person as image of God, that is, as centration of energies of love. The plasticity of culture in relation to nature—*natureculture*—can shape the material-discourse from which our academic rituals emerge and with which these rituals intra-act. In birthing a Catholic university that lives its catholicity—a wholemaking that contemplates the person as co-creative within the divine heart of matter—we continuously bring to birth the God who is Love.

The Christian imagination recreates this hope by which not just the human person but all creation is oriented. In this, its potential to draw into something new the diverse religious energies of creativity, Christianity as a religion of evolution is itself a technology of nature. If spirituality is the "focus on the stories and the myths of something more that go beyond the here and now and tell us what the here and now can become",[43] then the Christian incarnational imagination is itself a technology that offers the hope of envisioning and enacting a more humanizing and life-giving congruence between the sacred and secular than either the myth of Manifest Destiny that created "America" or the techno-meritocratic myths that delude us today. In the outworking and co-creating of this imagination lies hope for Christians and our Catholic institutions that we may image the very divine indefinability that seeks not our own ends and instead mirror a calling to live for something greater than ourselves for the rest of humanity. In a time of global pandemic and racialized violence, such is the hope of a Church and world in need of something more.

**Funding:** This research received no external funding.

**Institutional Review Board Statement:** Not applicable.

**Informed Consent Statement:** Not applicable.

**Conflicts of Interest:** The author declares no conflict of interest.

## Notes

[1] (Rude et al. 2015, p. 152). Among African Americans (the highest in affirming the promotion of racial understanding), Hispanics, Asians, and whites, students in all four groups were lower (on a four-point scale) at the end of their freshman year than upon arriving at college, and lower again by their senior year in the longitudinal study.

2  See for example, (Jack 2019, p. 22).

3  See for example (Delio 2018, chps. 2 and 3). Delio demonstrates how the Newtonian mechanics that gained ascendency during the 18th century still dominate our collective imaginations. Largely absent our cosmological framework are insights that are now over a century old, those emerging with Einstein's special theory of relativity in 1905. While Newton thought the material universe was made of inert matter, Einstein showed us that matter is simply a repackaged form of energy ($E = mc^2$). On the heels of Einstein came experiments, most notably the "double slit" that showed that particles exist simultaneously in multiple possible states (as both wave and particle in the case of light), thereby laying the groundwork for quantum physics. Particularly through the pioneering work of physicist Werner Heisenberg, the "radical turn of events" posed by quantum physics opens the door today to reverse the "holdover" effect of early modern mechanical philosophy—in Delio's words, the removal of mind from matter.

4  (Cobb 2015, p. 86). As we'll see below with the work of Karen Barad, the very act of observation produces physical reality. If the measure of a particle's position alters its momentum, ala Heisenberg's "uncertainty principle," there is then no determinism in nature, and there is no distinction between the act of observation and what is observed. See also (Delio 2018, pp. 19–20).

5  (Palmer and Zajonc 2010, pp. 127–28). Palmer echoes Keenan in recounting decades of work speaking to faculty about the "privatization of the professoriate" and the "fragmentation" that is encouraged by the structure of the academy into "silos called disciplines and departments."

6  I borrow the language of "minoritized" (rather than "minority") from, among others, educational researcher Shaun Harper who explains, "Persons are not born into a minority status nor are they minoritized in every social context . . . Instead they are rendered minorities in particular situations and institutional environments that sustain an overrresprentation of Whiteness." (Harper 2012).

7  See (Jack 2019). While focusing primarily on class differences among students, Jack notes that the "structural exclusion" of university policies that push poor students to the margins exacerbate class differences, "often in ways that connect to historical legacies of race and exclusion" (pp. 22–23). While this is true of students who are poor but come to college with an "early introduction" to collegiate life via a boarding or preparatory secondary school (the *privileged poor*), it's exacerbated for those who are both poor *and* unfamiliar with this new world," (the *doubly disadvantaged*), p. 11.

8  I bracket this term "white" for now. See below for an engagement with the critical race theory of George Yancy and Katie Grimes which problematizes an overly facile acceptance of what it means to be "white" in the contemporary American context.

9  (Delio 2020, p. 84). For a further introduction to transhumanism see for example (More 2013, pp. 3–17).

10  (Barad 2007). chapter 2. From the classical understanding of diffraction as the way waves combine when they overlap, Barad procedes to a quantuam understanding of the term and referecnes the two-slit diffraction experiement (cf. n. 3 above) as evidence for the superposition of matter (that under certain circumstances, particles exhibit wavelike behavior). Quoting Donna Haraway, Barad proposes that as "diffraction patterns record the history of interaction, interference, reinforcement, and difference . . . diffraction can be a metaphor for another kind of consciousness." (Dolphijn and van der Tuin 2012, p. 51).

11  Katie Walker Grimes develops this distinction from the "legacy" of slavery. See for example her "(Grimes 2017, pp. 41–60).

12  (Dolphijn and van der Tuin 2012, p. 69), both emphases added. See also (Barad 2007, chp. 8). Barad is careful to note that by entanglement, she does not simply mean "just any kind of connection, interweaving, or enmeshment in a complicated situation," (Barad 2007, p. 160). Rather, as above (n. 3 and 4), this term for the "the characteristic trait of quantum mechanics" points to the fact that physical reality is interconnected at the deepest levels we currently know. See also (Simmons 2014, p. 146).

13  For Barad, apparatuses, rather than mere tools or experimental simulations, are themselves phenomena. As such, "an apparatus is not premised on inherent divisions between the social and the scientific, the human and the nonhuman, nature and culture. Apparatuses are practices through which these divisions are constituted." (Barad 2007, pp. 169–70).

14  Introduced by Anibal Quijano, decoloniality encompasses multiple, contextual conceptualizations and actionings. It is "not a new paradigm or mode of critical thought," argue Walter Mignolo and Catherine Walsh, but "a way, option, standpoint, analytic, project, practice, and praxis." "Introduction" in (Mignolo and Walsh 2018, pp. 4–5).

15  Catherine E. Walsh, "Interculturality and Decoloniality" in (Mignolo and Walsh 2018, p. 76).

16  This term of the Guna people (today's North-western Colombia/South-eastern Panama) meaning "land in its full maturity" can be broadly understood as the Americas especially of South America in relation to the Caribbean. See "Introduction" in (Mignolo and Walsh 2018, p. 10).

17  Walsh, "The Decolonial *For*," in (Mignolo and Walsh 2018, p. 24).

18  Mignolo, "What Does It Mean to Decolonize?" in (Mignolo and Walsh 2018, p. 119). Mignolo offers a genealogical uncovering of the origins of the idea of progress as popularly understood today as emerging in the late (Western) medieval period, arguing that today, the West's "particular ontology of history continues to assert its universality." *Nonmodern*, he argues, is a flexible and necessary concept "to illuminate the coexistence of temporalities and modes of living and thinking that are neither premodern nor postmodern," p. 117.

19  (Barad 2007, p. 180). Holding richer possibilities for imagining *space* than our geometrically/Newtonian-inspired concern with shapes and sizes, *topology*, Barad notes, investigates questions of connectivity and boundaries. Thus, spatiality is intra-actively

produced, and, advancing Einstein's special theory of relativity (by which the couple "space" and "time" become the single "space-time"), she proposes speaking of "spacetimemattering" as "an agential realist term that acknowledges the iterative materialization of space-time-matter." See Barad, "What Flashes Up: Theological-Political-Scientific Fragments" in (Keller and Rubenstein 2017, p. 78, n.24).

20  Mignolo, "Colonial/Imperial Differences," in (Mignolo and Walsh 2018, pp. 183–84).

21  I borrow from Carmelite William McNamara's definition of contemplation as a "long, loving look at the real."

22  Using data from the Center on Education and the Workforce, Carnevale and Strohl note that the "top tier" 468 colleges are those "most-, highly-, and very-competitive" schools which admit, respectively, less than 1/3, between 1/3 and 1/2, and 1/2–3/4 of applicants (45). "Open access" colleges are those from the bottom two categories of selectivity, "less- and non-competitive schools," whose rates of acceptance are 85% or higher. Carnevale and Strohl demonstrate how, since 1995, as overall African American and Hispanic freshmen enrollments have increased, so too have the number of seats occupied by students in these groups in open access schools. Furthermore, controlling for student preparation and other "personal characteristics," a little less than half of racially minoritized students do not complete a Bachelor's degrees due to lack of resources and other forms of support (33).

23  Concentrating their findings on African Americans and Hispanics, Carnevale and Strohl demonstrate how spatial and social isolation (in poorer neighborhoods) from the general society points to race as giving "additional power to the negative effects of low-income status" and limiting "the positive effects of income gains, better schools, and other educational improvements," 12.

24  Andrew Prevot, "Sources of a Black Self? Ethics of Authenticity in an Era of Anti-Blackness" in (Lloyd and Prevot 2017, p. 80).

25  Lauren Berlant in (Yancy 2017). First quoting Berlant, Yancy argues, "'Cruel optimism' names a relation of attachment to compromised conditions of possibility.' So, one might say that there is a desire for robust democratic inclusion, the desire for the recognition of Black humanity, but such desires take place within the relational context of a form of white concession that these will never be achieved or achievable. On this score, whiteness 'gives,' but only enough to keep hope in place," pp. 120–21.

26  In her work on white fragility, Robin DiAngelo helps to unveil how strategies and practices of "niceness" actually creation a culture "in which white people assume that niceness is the answer to racial inequality and people of color are required to maintain white comfort in order to survive." See (DiAngelo 2019).

27  See (Patterson 1982, p. 7). It is this "loss of ties to birth in both ascending and descending generations" that especially resonates with the "disconnection" thesis of bell hooks, above, and, thus, a central theme of our current discussion.

28  Katie Grimes, in conversation with Yancy, argues, "We will not miscegenate or immigrate our way out of anti-blackness supremacy." In other words, the "browning of America" with new waves of nonwhite migrants, rather than signaling a break with the nation's racial past, will further define the color line as between black and nonblack. Who counts as "white" is judged by the standard of a moving goalpost. See her "Black Exceptionalism: Anti-Blackness Supremacy in the Afterlife of Slavery" in (Lloyd and Prevot 2017, pp. 58–60).

29  In the words of theologian Paul Tillich. (Delio 2020, p. 167).

30  (Copeland 2009, p. 51). On this system of Black concubinage, see her "Introduction."

31  Copeland asserts that, "Our public memory as a nation suppresses the depth of our entanglement in racial slavery" and, therefore, "the dangerous memory of the messianic God" (J.B. Metz) offers a praxical chance for authenticity in which (in line with the diffractive methodology of an agential realism) "past and present compenetrate each other, for the past is never a fixed reality." (Copeland 2018, pp. 83, 96–99).

32  (Francis 2015, n. 111). Emphasis added.

33  For more on the nature of solidarity, see (Beyer 2010, p. 159, n. 66). See also (Francis 2015) in which the Pope calls for both an intra- and intergenerational solidarity (nos. 158–62), a point much needed in considering, as Carnevale and Strohl do above, the intergenerational nexus of race and poverty in the United States.

34  (Delio 2020, p. 131). From the Greek for "the art of steering," cybernetics is about circular-causal relationships, whereby the action of a complex dynamical system generates a change in the environment, which in turn prompts a change in the system. See (Delio 2020, pp. 70–71).

35  On *techne* as the act of "bringing forth" and its relation to *poesis*, the art of making something out of existing materials," see (Delio 2020, p. xv).

36  Noting that Black faculty and staff members are the primary reason why Black students stay at majority-white institutions, Ebony O. McGee lists counting service work toward promotion among her 12 ways that "white faculty members can better support Black academics in their department and across campus." See her (McGee 2020).

37  (Fitzpatrick 2019, p. 227). Emphasis added.

38  See for example (Palmer 2007).

39  See "Introduction" in (Mignolo and Walsh 2018, p. 10).

40  Barad, "What Flashes Up: Theological-Political-Scientific Fragments" in (Keller and Rubenstein 2017, p. 75).

41  For example, via a "pedagogy of liminal mutuality." See (Mayer 2019).

42    Teresa A. Nance, Vice President for Diversity, Equity, and Inclusion, Villanova University. Personal correspondence used with permission.

43    Philip Hefner quoting Mihaly Csikszentmihalyi in (Hefner 2002).

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
