# Peer review of "Re-Membering Catholicity: Higher Education, Racial Justice, and the Spirituality of the Posthuman University"

_religions, doi:10.3390/rel12080645_

Round 1
Reviewer 1 Report
See attached document.

Reviewer 2 Report
Wide-ranging pluridisciplinary method of presentation;
arguments hang together;
As reader, I felt drawn by the highly critical approach, awaiting with expectation author's proposed solution;
not disappointed when author anchors solution on conversion
Only Problem: Language not easily accessible; not for the general public;
But this is an important and useful read for Catholic education
Author Response
Dear Reviewer,
Thank you for your comments and suggestions on my work. I especially appreciate the affirmation of the pluridisciplinarity of my presentation, which was a deliberate approach in this project in keeping with the thesis of the paper. On the language of the paper as not easily accessible for the general public, I am grateful for this observation. It is one that I often level as a self-critique of my work. While not “on my radar” in initially approaching this particular project, I will take your observation and further incorporate it in future projects that hopefully extend my contributions in the field of Catholic education to a more general reading audience. Thank you again for your time and gracious comments!